# Continually Improving Extractive QA via Human Feedback

**Ge Gao**◇*, **Hung-Ting Chen**♣*, **Yoav Artzi**◇, and **Eunsol Choi**♣
◇Department of Computer Science and Cornell Tech, Cornell University
♣Department of Computer Science, The University of Texas at Austin
{ggao, yoav}@cs.cornell.edu {hungtingchen, eunsol}@utexas.edu

## Abstract

We study continually improving an extractive question answering (QA) system via human user feedback. We design and deploy an iterative approach, where information-seeking users ask questions, receive model-predicted answers, and provide feedback. We conduct experiments involving thousands of user interactions under diverse setups to broaden the understanding of learning from feedback over time. Our experiments show effective improvement from user feedback of extractive QA models over time across different data regimes, including significant potential for domain adaptation.

## 1 Introduction

The deployment of natural language processing (NLP) systems creates ample opportunities to learn from interaction with users, who can often provide feedback on the quality of the system output. Such feedback can be more affordable and abundant than annotations provided by trained experts. It can also be obtained throughout the system's lifetime, opening up opportunities for continual learning,[1] where the system improves over time, and evolves as the world and user preferences change.

The combination of human feedback and continual learning presents exciting prospects, but is relatively understudied, partially because of the challenges it poses. Mainly because it requires deploying models with human users over many interactions for both development and evaluation, for example to study the interplay between the learning signal and model design, the impact of the initial model, and learning progression over time.

Focusing on extractive question answering (QA), we study iteratively improving an NLP system by learning from human user feedback over time. We

---

*Equal contribution.

[1]The term *continual learning* is at times used to refer to a scenario where models adapt to new tasks over time. We study improving the model continually on its original task.

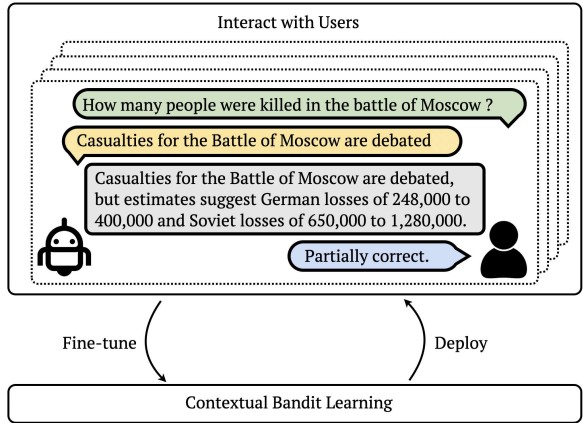

Figure 1: Illustration of our setup. We alternate between interaction and learning phases. At the interaction phase, given a user question and context paragraph, the model predicts the question is unanswerable or a span in the paragraph as the answer. The human user provides "correct", "partially correct" or "wrong" feedback to the model answer by validating the answer in context. The system aggregates user feedback, and the model learns from feedback data at the learning phase by updating its parameters, with the goal of improving over time.

create and deploy an information-seeking scenario, where users pose a question, receive a prediction from a model which aims to answer the query given a single evidence document, and provide feedback. The system iteratively improves over rounds, where each round is made of deploying the system to interact with users, followed learning. Figure 1 illustrates our setup and learning paradigm. This scenario has been studied for embodied agents in constrained environments (Thomason et al., 2015; Kojima et al., 2021; Suhr and Artzi, 2022), but QA proposes new challenges for using such interaction data, for example in its significantly richer lexical space. To the best of our knowledge, ours is the first study to show a QA system improving via interaction with human users over time.

Our setup is designed to study practical interaction with users. Following prior work on collecting information-seeking queries (Choi et al., 2018; Clark et al., 2020), we prompt users to ask ques-

tions that they do not know the answers to. This naturally elicits questions that cannot be answered under the fixed-evidence QA scenario (Kwiatkowski et al., 2019). We observe that the presence of unanswerable questions dramatically impacts learning, making the process sensitive to the model's calibration on answerability. We address this by modeling question answerability separately from span extraction: our model first predicts if the question is answerable and only then extracts an answer span.

We use a simple three-option feedback signal ("correct", "partially correct", or "wrong"). While less informative than more complex cues, such as natural language, such feedback imposes little interaction overhead. We map the feedback signal to manually-specified reward values, and formulate learning as a contextual bandit problem.

We conduct multiple studies, focusing on low data regimes, with the aim of demonstrating robustness to challenging scenarios where only a small fraction of users provide feedback, and rapid improvement is necessary with little feedback. Over nine rounds of deployment with over 1,800 user interactions, our approach shows an overall improvement from 39 to 67 F1 score on our newly collected Wikipedia-based data. We conduct extensive studies of our approach, including showing effective domain adaptation from a different source domain that does not even support unanswerable questions. Taken together, our results demonstrate the potential of NLP systems to learn from user feedback over time, specifically for extractive QA. Our data and codebase are publicly available at https://github.com/lil-lab/qa-from-hf.

## 2 Related Work

Our use of human feedback is related to work recently-termed reinforcement learning from human feedback (RLHF; e.g., Ziegler et al., 2019; Stiennon et al., 2020; Nakano et al., 2021; Ouyang et al., 2022; Scheurer et al., 2023). Largely, these methods rely on soliciting pair-wise comparisons from annotators, which are used to train a reward model to be used in an RL process. We adopt a different approach: soliciting feedback from users on single outputs to their queries, and mapping the feedback to reward values to be used in offline contextual bandit learning. An important consideration motivating our choice is that pair-wise comparison, although suitable for paid annotators, is less suitable for soliciting feedback from actual users. Head-to-head comparison between learning a reward model and directly mapping feedback to reward values, as we do, remains an important direction for future work. Human feedback has also been studied without RL (Xu et al., 2022; Thoppilan et al., 2022). A critical distinction of our work is the focus on continual learning (i.e., iteratively running our process over many rounds) and the dynamics it creates, whereas the work mentioned above (both using and not using RL) focused on improving models a single time.[2]

Learning from explicit or implicit feedback for language tasks was studied beyond recent interest in RLHF, including for machine translation (Nguyen et al., 2017; Kreutzer et al., 2018b,a), semantic parsing (Artzi and Zettlemoyer, 2011; Lawrence and Riezler, 2018), question answering (Gao et al., 2022), and chatbots (Jaques et al., 2020). Similar to the work mentioned earlier, this line of work did not explore iterative continual learning, as we emphasize. The iterative process was studied in the context of embodied instruction generation (Kojima et al., 2021) and following (Thomason et al., 2015; Suhr and Artzi, 2022). In contrast, we study QA on a wide range of Wikipedia articles, data with high lexical diversity. Others obtained complete labels as feedback (Iyer et al., 2017; Wang et al., 2016), a process with higher overhead. RL using human feedback has also been studied for non-language problems (e.g., Knox and Stone, 2015; Warnell et al., 2017; MacGlashan et al., 2017; Christiano et al., 2017).

Prior work on learning from interaction for QA used synthetic interactions and feedback simulated from supervised data (Campos et al., 2020; Gao et al., 2022). We study human feedback from information-seeking human users who frequently pose challenging questions which may be unanswerable from the given document. Li et al. (2022) proposed a process that involves crowdworkers providing rating and explanation to given for given question-answer pairs to improve a QA model post-deployment. They control data quality with manual reviewing and multiple annotations. Our setup has lower interaction overhead, at the cost of providing a less informative, and at times noisy signal. We also go beyond one-time improvement in studying iterative deployment, providing insight into how the model improves over time.

---

[2]Ziegler et al. (2019) note that RLHF can be executed iteratively, but do not report the details or analysis of doing so.

## 3 Interaction Scenario

We focus on the task of extractive QA, which creates an intuitive feedback solicitation scenario. It is relatively easy to visualize the model output (i.e., a span of text) in the context it is extracted from (i.e., the evidence paragraph), for the user to verify the answer. Automated evaluation is also well understood (Bulian et al., 2022), allowing reliable measurement of system performance over time.

We deploy our QA system in rounds. Each round starts with an interaction phase, followed by a learning phase. At the interaction phase, users interact with a fixed, deployed model and provide feedback. We aggregate this interaction data until we collect a fixed amount of feedback data to enter a learning phase. The feedback is collected during natural user interactions (i.e., the model is deployed to fulfil its task of answering user questions). At the learning phase, we update the model parameters based on the aggregated feedback data. Because we observe no new feedback data during the learning phase, this creates an offline learning scenario (Levine et al., 2020),[3] which is practical for deployed systems. Except data aggregation, it requires no integration of the learning process into the deployed interactive system. The separation between deployment and training also enables sanity checks before deploying a new model, and for hyperparameter tuning as in supervised learning.

Each interaction starts with a user posing a question. The model computes an answer, and returns it to the user alongside a visualization of it in the context text from which it was extracted. The user provides feedback, by selecting one of three options: "correct", "partially correct" or "wrong". Table 1 shows examples from our studies.

Formally, let a question $\bar{q}$ be a sequence of $m$ tokens $\langle q_1, \ldots, q_m \rangle$ and a context text $\bar{c}$ be a sequence of $n$ tokens $\langle c_1, \ldots, c_n \rangle$. A QA model at round $\rho$ parameterized by $\theta_\rho$ computes two probability distributions: a binary distribution indicating if the question is answerable or not by the context text $P_u(u|\bar{q}, \bar{c}; \theta_\rho)$, where $u \in \{\text{ANS}, \text{UNANS}\}$; and a distribution over answer spans in the context text $P_s(i, j|\bar{q}, \bar{c}; \theta_\rho)$, where $i, j \in [1, n]$ and $i \leq j$. If $\arg\max_{u \in \{\text{ANS,UNANS}\}} P_u(u|\bar{q}, \bar{c}; \theta_\rho) = \text{UNANS}$, the model returns to the user that the question is unanswerable. Otherwise, the model returns the highest probability span $\arg\max_{i,j} P_s(i, j|\bar{q}, \bar{c}; \theta_\rho)$. Given

---

Question: What did Saladin die from?
Answer: a fever
Context: Saladin died of a fever on 4 March 1193 (27 Safar 589 AH) at Damascus ...
Feedback: Correct

Question: Where does the name St Albans come from?
Answer: Alban
Context: St Albans takes its name from the first British saint, Alban. The most elaborate version of his story, Bede's Ecclesiastical History of ...
Feedback: Partially Correct

Question: In Hindu mythology what were the names of Radha's lovers?
Answer: [Unanswerable given the context paragraph]
Context: Radha in her human form is revered as the milk-maid (gopi) of Vrindavan who became the beloved of Krishna. One of ...
Feedback: Partially Correct

Question: What percentage of people in Reno are White?
Answer: [Unanswerable given the context paragraph]
Context: As of the census of 2010, there were ... The city's racial makeup was 74.2% White, 2.9% African American, 1.3% Native American, 6.3% Asian, 0.7% Pacific Islander, 10.5% some other race, and 4.2% ...
Feedback: Wrong

Table 1: User interaction examples: each example is composed of user question, context paragraph, model-predicted answer, and user feedback. Appendix C lists additional examples.

---

the model output, the user provides feedback $f \in \{\text{CORRECT}, \text{PARTIALLY-CORRECT}, \text{WRONG}\}$. Each user interaction generates a data tuple $(\bar{q}, \bar{c}, \hat{u}, \hat{i}, \hat{j}, f, \theta_\rho)$, where $\hat{u}$ is the binary answerability classification, and $\hat{i}$ and $\hat{j}$ are the start and end indices of the extracted answer, if the question is classified as answerable.

## 4 Method

We initialize the model with supervised data, and improve it by learning from user feedback through an offline contextual bandit learning process.

### 4.1 Model and Initialization

We use a standard BERT-style architecture (Devlin et al., 2019). The input to the model is a concatenation of the question $\bar{q}$ and the context text $\bar{c}$. We separately classify over the context tokens for the answer span start and end to compute the span distribution $P_s$ (Seo et al., 2017), and compute the binary answerability distribution $P_u$ with a classification head on the CLS token (Liu et al., 2019).

We initialize the model parameters with DeBERTaV3 weights (He et al., 2023),[4] and fine-tune us-

---

[3]Our overall iterative process can also be seen as batched contextual bandit (Perchet et al., 2016).

[4]We use the Hugging Face (Wolf et al., 2020) version of DeBERTaV3 (He et al., 2023): *microsoft/deberta-v3-base*.

ing supervised data to get our initial model. This is critical to get a tolerable experience to early users. We usually use a small number of examples ($\leq 512$ examples), except when studying domain transfer. The training loss sums over the three cross-entropy classification losses, with a coefficient $\lambda$ to weigh the binary answerable classification term.

## 4.2 Bandit Learning

We learn through iterative deployment rounds. In each round $\rho$, we first deploy our model to interact with users (Section 3) and then fine-tune it using the data aggregated during the interactions. Each user interaction generates a tuple $(\bar{q}, \bar{c}, \hat{u}, \hat{i}, \hat{j}, f, \theta_\rho)$, where $\bar{q}$ is a question, $\bar{c}$ is a context text, $\hat{u}$ is the answerability classification decision, $\hat{i}$ and $\hat{j}$ are the returned span boundaries if a span was returned, and $\theta_\rho$ are the model parameters when the interaction took place.

**Policy** We formulate a policy that casts answer prediction as generating a sequence of one or two actions, given a question $\bar{q}$ and a context $\bar{c}$. This sequential decision process formulation, together with the multi-head model architecture, allow to control the losses of the different classification heads by assigning separate rewards to the answerability classification and the span extraction. The policy action space includes classifying if the question is answerable (ANS) or not (UNANS) and actions for predicting any possible answer span $[i, j]$ in $\bar{c}$.

The set of possible action sequences is constrained. At the start of an episode, the policy first predicts if the question is answerable or not. The probability of the action $a \in \{\text{ANS}, \text{UNANS}\}$ is $P_u(a|\bar{q}, \bar{c}; \theta)$. Span prediction action are not possible, so their probabilities are set to 0. If the UNANS action is selected, the episode terminates. Otherwise, the second action selects a span $[i, j]$ from $\bar{c}$ as an answer, and the episode terminates. The probability of each span selection action is $P_s(i, j|\bar{q}, \bar{c}; \theta)$. Answerability prediction actions are not possible, so their probabilities are set to 0.

**Reward Values** We do not have access to a reward function. Instead, we map the user feedback $f$ to a reward value depending on the action (Table 2), and cannot compute rewards for actions not observed during the interaction. The policy formulation, which casts the prediction problem as a sequence of up to two actions, allows to assign different rewards to answerability classification and

| Action | Feedback | | |
| | CORRECT | PARTIALLY-CORRECT | WRONG |
| --- | --- | --- | --- |
| UNANS | 1 | 0 | -1 |
| ANS | 1 | 1 | 0 |
| $[\hat{i}, \hat{j}]$ | 1 | 0.5 | -0.1 |

Table 2: The mapping of user feedback to specific actions to reward values. $[\hat{i}, \hat{j}]$ is the span predicted during the interaction, if one is predicted.

span extraction. For example, if we get WRONG feedback when an answer is given, we cannot tell if the answerability classification was correct or not. Our formulation allows us to set the reward value of the first action to zero in such cases, thereby zeroing the answerability classification loss. The reward values were determined through pilot studies. For example, we observed that models overpredict unanswerable, so we set a relatively large penalty of -1 for wrongly predicting unanswerable.

**Learning Objective** We use a policy gradient REINFORCE (Williams, 1992) objective with a clipped inverse propensity score coefficient (IPS; Horvitz and Thompson, 1952; Gao et al., 2022) and an entropy term for the answerability binary classification. IPS de-biases the offline data (Bietti et al., 2021), and also prevents unbounded negative loss terms (Kojima et al., 2021). The entropy term regularizes the learning (Williams, 1992; Mnih et al., 2016). If we substitute the policy terms with the predicted model distributions, the gradient for an answerable example with two actions with respect to the model parameters $\theta$ is:

$$\nabla_\theta \mathcal{L} = \alpha_1 r_1 \nabla P_u(\hat{u}|\bar{q}, \bar{c}; \theta) \qquad (1)$$
$$+ \alpha_2 r_2 \nabla P_s(\hat{i}, \hat{j}|\bar{q}, \bar{c}; \theta) + \gamma \nabla H(P_u(\cdot|\bar{q}, \bar{c}; \theta))$$
$$\alpha_1 = \frac{P_u(\hat{u}|\bar{q}, \bar{c}; \theta)}{P_u(\hat{u}|\bar{q}, \bar{c}; \theta_\rho)} \; ; \; \alpha_2 = \frac{P_s(\hat{i}, \hat{j}|\bar{q}, \bar{c}; \theta)}{P_s(\hat{i}, \hat{j}|\bar{q}, \bar{c}; \theta_\rho)} \; ,$$

where the $\alpha_1$ and $\alpha_2$ are IPS coefficients for the first (answerability classification) and second (span extraction) actions, $r_1$ and $r_2$ are the corresponding reward values, $\gamma$ is a hyperparameter, and $H(\cdot)$ is the entropy function. For examples the model predict as unanswerable, the second term is omitted.

**Deployment and Learning Process** Algorithm 1 outlines our process. Each round (Line 2) includes interaction (Lines 4–14) and learning (Lines 15–18) phases. During interaction, given a question and context (Line 5), we classify if it is answerable in the given context (Line 7) and extract the answer span (Line 8). Depending on the classification, we

**Algorithm 1** Deployment and Learning.

1: $\mathcal{D} \leftarrow \emptyset$
2: **for** round $\rho = 1 \cdots$ **do**
3: $\quad D_\rho \leftarrow \emptyset$
4: $\quad$ **for** interaction $t = 1 \cdots T$ **do**
5: $\qquad$ Observe a question $\bar{q}^{(t)}$ and context $\bar{c}^{(t)}$
6: $\qquad$ Predict if answerable and answer span:
7: $\qquad\quad \hat{u}^{(t)} \leftarrow \arg\max_u P_u(u|\bar{q}, \bar{c}; \theta_\rho)$
8: $\qquad\quad \hat{i}^{(t)}, \hat{j}^{(t)} \leftarrow \arg\max_{i,j} P_s(i, j|\bar{q}, \bar{c}; \theta_\rho)$
9: $\qquad$ **if** $\hat{u}^{(t)} =$ ANS **then**
10: $\qquad\quad$ Display the span $[\hat{i}^{(t)}, \hat{j}^{(t)}]$ from $\bar{c}^{(t)}$ as answer
11: $\qquad$ **else**
12: $\qquad\quad$ Display that the question is not answerable
13: $\qquad$ Observe user feedback $f^{(t)}$
14: $\qquad D_\rho \leftarrow D_\rho \cup \{(\bar{q}^{(t)}, \bar{c}^{(t)}, \hat{u}^{(t)}, \hat{i}^{(t)}, \hat{j}^{(t)}, f^{(t)}, \theta_\rho)\}$
15: $\quad$ **for** $e = 1 \cdots E$ epochs **do**
16: $\qquad$ **for** $D' = \frac{B}{2}$ examples from $D_\rho$ **do**
17: $\qquad\quad D' \leftarrow D' \cup \{$ sample $\frac{B}{2}$ examples from $\mathcal{D}\}$
18: $\qquad\quad$ Update model parameters $\theta$ using $D'$ with the gradient in Equation 1
19: $\quad \mathcal{D} \leftarrow \mathcal{D} \cup D_\rho$

either display the answer (Line 10) or return that the question is not answerable in the given context (Line 12), and solicit feedback (Line 13). We aggregate the interaction data over time (Line 14). During learning, we use rehearsal (Rebuffi et al., 2017) for each update, creating a batch of size $B$ by mixing examples from the most recent interactions (Line 16) and previous rounds (Line 17) to update the model parameters (Line 18).

## 5  Experimental Setup

**Deployment Setup**  We study our approach using Amazon Mechanical Turk. We use Wikipedia data, a relatively general domain, thereby making the data we observe lexically rich and topically diverse. Our interaction scenario is designed to elicit information-seeking behavior. Following prior work (Choi et al., 2018), users are not given the context text when they pose the question, and are instructed to ask questions they do not know the answer to. This results in a significant number of questions that cannot be answered given the evidence context. To increase user information-seeking engagement, we do not dictate the topic to ask about, but allow users to select a topic (e.g., "Saladin") and an aspect (e.g., "Death") from a set that is generated randomly for each interaction (Eisenschlos et al., 2021).[5]

Each topic-aspect pair is associated with an evidence paragraph, which serves as context to the

model. We extract topics, aspects, and evidence paragraphs from Wikipedia.[6] We focus on learning from natural user feedback, so we do not perform posthoc filtering of noise beyond removing adversarial crowdworkers (e.g., workers who always ask the same question).[7] Appendix B provides more details about our interface and data collection. Appendix A gives training implementation details (e.g., hyperparameters).

**Evaluation Data**  It is critical for test data to come from the distribution the users create while interacting with the system for the evaluation to reflect the system performance. This makes existing benchmarks of lesser relevance to our evaluation. We collect a held-out testing set that matches the observed distribution by interleaving the process in our deployment. We design a data annotation procedure based on our regular interaction task, and randomly opt to ask workers to provide an answer to the question they just posed rather than showing them the model answer and soliciting their feedback. This branching in the process happens only after they ask the question, and workers see no indication for it when posing the question, so the questions we collect this way follow the same distribution the systems see. We collect around 100 such annotated examples each round during each of our experiments. This is a small set, but unified across rounds within each experiment, it results in a set large enough for reliable testing. To increase the quality of the annotation, we separately collect two additional annotations for each such test question. Appendix B provides further details on the test data collection and annotation.

We also evaluate our approach on the English portion of TyDiQA (Clark et al., 2020) development set (954 examples, excluding Yes/No questions), which was collected in a setting relatively similar to our evaluation data. Each example includes a question, a context paragraph, and a set of three human reference annotations.

**Evaluation Metrics**  We focus on how different measures develop throughout the system's deploy-

---

[5]We concatenate the topic and aspect to the evidence paragraph as input to the model, but do not allow selecting these as answer spans.

[6]Our pool of topics is a subset of previously compiled popular entities (Onoe et al., 2022) based on the number of contributors and backlinks. Aspects of each topic correspond to section titles on its Wikipedia page. Each topic in our pool contains at least four sections, with each section shorter than 490 tokens (excluding subsections).

[7]Qualitative analysis of 100 randomly sampled examples from the first round of our long-term study (Section 6.1) shows a feedback noise rate of 18%.

ment. We compute deployment statistics on the aggregated interactions, including answerability prediction, feedback, and the reward values it maps to. We use the testing data to compute various testing statistics on each deployed model, including token F1 and answerability classification accuracy. For both datasets, we follow prior work (Clark et al., 2020) to consolidate the annotations. When the annotators disagree on the answerability of the question, we take the majority vote as the ground truth. We compute token-level F1 against the three annotations by considering the reference answer that will give the highest F1 score.

## 6 Results and Analysis

We conduct two deployment studies: a long-term deployment to study the effectiveness and dynamics of the process (Section 6.1), and a shorter-term study to evaluate the impact of certain design choices (Section 6.2). We also evaluate the difference between iterative rounds and a one-time improvement (Section 6.3). Section 6.4 provides an additional experiment analyzing the sensitivity of our learning process to the observed data.

### 6.1 Long-Term Experiment

We deploy for nine rounds of interaction and learning, starting from an initial model trained on 512 SQuAD2 examples. Each round of interaction includes around 200 interactions with feedback. We observe 1,831 interactions over the nine rounds. We also concurrently collect about 100 test examples per round, a total of 869 examples.[8] The total cost of this study is 3,330USD.

Figure 2 shows statistics from the nine rounds of deployment. The frequency of "correct" feedback increases from 47.71% during the first round to 69.35% at the last round, with "wrong" feedback decreasing (40.82→16.08%). The frequency of "partially correct" remains stable.While the trend is generally stable, we see that temporary dips in performance occur (round 3 and rounds 5–7), even though learning can recover from them. Reward trends are similar, with the total reward increasing (1.03→1.29), and both answerability classification (0.58→0.77) and answer span (0.48→0.74) rewards increasing.

Answerability prediction rates shed light on the learning dynamics, especially early on, when we

---

8Exact numbers per round vary slightly because of how workers capture tasks on Amazon Mechanical Turk.

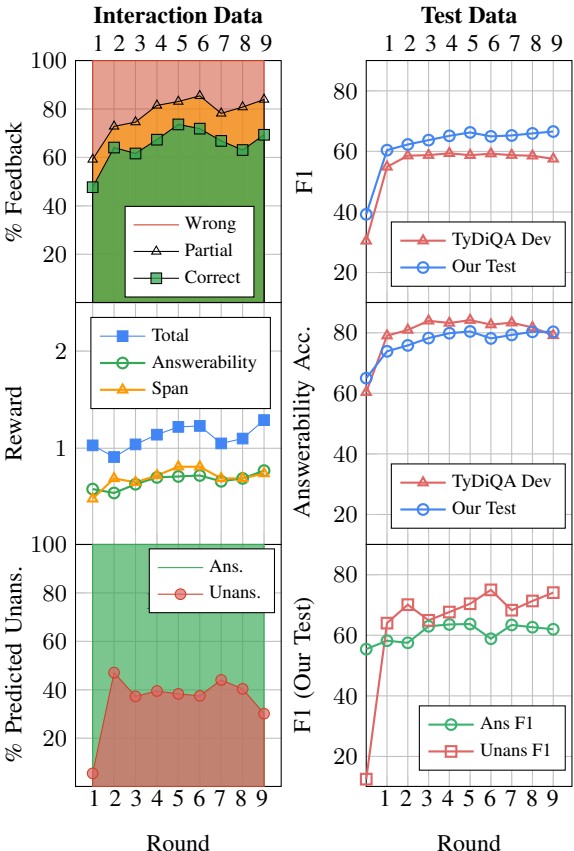

Figure 2: Statistics for the interaction data and the model performance on test sets after training on interaction data from each round. *Our test* set is the union of all test data collected during the nine rounds.

observe dramatic changes. The initial model under-predicts questions as unanswerable, as can be seen by comparing prediction rates in the first round (5.5%) to the ratio of unanswerable questions in the test data (stable at 35–40%). This is followed by a dramatic correction: the model slightly over-predicts unanswerable in Round 2 (42%). Starting from Round 3, the model stabilizes to predict around 38% questions as unanswerable.[9]

Test performance on our concurrently collected test data shows a similar positive trend, with both F1 (39.22→66.56) and answerability classification accuracy (65.02→80.32%) increasing over the nine rounds. TyDiQA performance increases initially, but shows a slight drop later on. The early improvements using our feedback data indicate similarity between TyDiQA and our user questions. However, the later decrease shows TyDiQA is still somewhat different, so as the model specializes given more

---

[9]Figure 6 and Figure 7 in the appendix show the statistics on the answerable and unanswerable subsets partitioned based on model prediction in the feedback data.

|         | CORRECT | PARTIALLY-CORRECT | WRONG |
|---------|---------|-------------------|-------|
| Round 1 | 47.71   | 11.47             | 40.83 |
| Round 9 | 47.85   | 14.35             | 37.80 |

Table 3: User adaptation study: feedback statistics to the initial model deployed at Round 1 and Round 9.

feedback data, it is expected to see a drop on Ty-DiQA, from which we observe no training signal. This difference is evident, for example, in the rate of answerable questions. Our test data maintains a rate of about 35–40%, while 50% of the examples in the TyDiQA data are unanswerable.

Further breaking down F1 performance based on ground-truth answerability shows that the initial model starts with non-trivial answerable F1 and improves gradually (55.43→62). But, it starts with very low performance on the unanswerable subset (12.5). It improves over time on the unanswerable subset as well (12.5→74.09), with a rapid improvement at the very first round (12.5→64.02).

User adaptation is a potential confounding factor, where performance trends might be explained by users changing their behavior (e.g., by simplifying their questions) while the model does not actually change its performance. We quantify user adaptation by deploying the Round 1 (parameterized by $\theta_1$) model alongside the final model at Round 9, for the same number of interactions. Interactions are assigned one of the models randomly with no indication to the user. Table 3 shows the feedback comparison for the two models. Except a small difference in "partially correct" and "wrong" feedback, we observe no change in the initial model performance, indicating user adaptation does not explain the performance gain.

## 6.2 Analysis on Model Variants

We study the impact of various design decisions by concurrently deploying five systems in a randomized experiment, including re-deploying our default configuration. Repeatedly deploying the default configuration serves as a fair comparison with other variants, being launched concurrently and sharing the same user pool.

The different variants copy the design decisions of the default configuration, unless specified otherwise. Each interaction is assigned with one of the systems randomly. Unless specified otherwise, each system aggregates around 200 interactions per round. We also concurrently collect a test set of around 200 examples per round.[8] This exper-

iment includes a total of 5,079 interactions, at a total cost of 6572.88USD. Figure 3 summarizes deployment and test statistics for this experiment, showing test results on the test data concurrently collected during this experiment. The reproduction of the default setup shows similar improvement trends to the long-term experiment (Section 6.1). We experiment with four variations:

**Weaker Initial Model**  We deploy a weaker initial model trained with a quarter of the training data (128 examples). This variant shows effective learning, with significant increase in "correct" feedback (26.67→55.61%) over the five rounds. However, it still lags significantly behind the default setup with its stronger initialization.

**Fewer Interactions Per Round (Fewer Ex.)**  We experiment with updating the model more frequently, with each round including around 100 examples instead of 200. We deploy this system for 10 rounds, so the total number of interactions it observes is identical.[10] This model performance is on par with that of the default setup. However, the percentage of unanswerable examples is slightly less stable than others, likely due to the larger variance in data samples.

**Answerability Classification (No CLS)**  We ablate using a separate answerability classification head. Instead, this model always select a span, either predicting an answer span in the context or indicating unanswerable given the context paragraph by predicting the CLS token (Devlin et al., 2019). We use a reward value mapping that gives a reward of 1 for CORRECT, 0.5 for PARTIALLY-CORRECT, and -0.1 for WRONG. This model over-predicts unanswerable in the first round, and afterwards keeps showing a lower classification accuracy compared to the default setup. Further analysis shows that the final model predicts 29% unanswerable on the ground-truth answerable subset, and only 65% unanswerable on the ground-truth unanswerable subset, significantly worse than the final model under default setup (25% and 80%). This indicates explicitly learning a separate head for answerability prediction is crucial.

**Domain Adaptation (NewsQA Initial)**  We train an initial model on the complete NewsQA training

---

[10]The first 5 rounds are deployed concurrently with other variants, and the last 5 rounds are deployed alone. We do not collect test examples alongside the later 5 rounds.

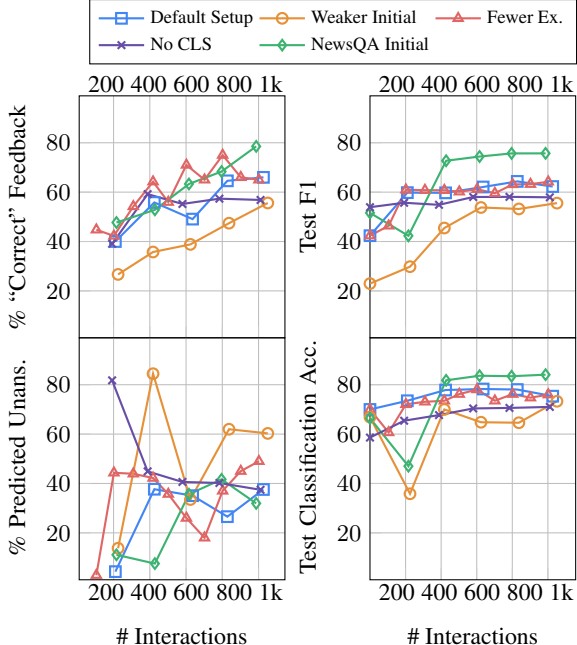

Figure 3: Interaction statistics (left) and model test performance (right) in the short-term experiment. We report test F1 and classification accuracy on the full test data collected across five rounds of deploying those models concurrently.

| | F1 | Ans. F1 | Unans. F1 | % Unans. |
|---|---|---|---|---|
| 1-Round | 56.57 | **60.73** | 49.70 | 24.97 |
| 4-Round | **64.34** | 56.93 | **79.01** | **41.41** |
| 8-Round | 63.13 | 56.98 | 75.31 | 40.79 |

Table 4: One-round vs. multi-round: we compare (from the top) the model trained on around 800 examples in one round, the model trained with four rounds of iterative improvement (around 200 ex. per round), and the model with 8-round of improvements (around 100 ex. per round). We report overall F1, F1 on the ground-truth answerable subset, F1 on the ground-truth unanswerable subset, and the percentage of model-predicted unanswerable examples.

dataset, a dataset with context texts that are significantly different than ours. The initial model is trained *without* a classification head, since NewsQA only contains answerable examples. We add a randomly initialized answerability classification head, and randomly output "unanswerable" for 10% of questions during Round 1, so that we observe feedback for this classification option. The model initialized with NewsQA shows promising performance, as previously indicated in simulated studies (Gao et al., 2022). In fact, it is the best-performing model in this study. We hypothesize that the model benefits from the larger initial training set, despite the domain shift, and only requires relatively small tuning for the new domain.

## 6.3 One-round vs. Multi-round

Iterative rounds of deployment and learning is core to our approach design. An alternative would be to collect all the feedback data with the initial model, and then update the model once. We compare our design to this alternative.

Concurrent to the Round 1 deployment in Section 6.2, we collect additional 500 feedback interactions from the 512-SQuAD2 initial model, and create a set of 800 interactions together with 100 examples from the experiment on fewer-interactions-per-round experiment and 200 examples from the

default experiment at the first round. We fine-tune the initial model with this set, essentially doing a single round with 800 examples, a quantity that other models only see across multiple rounds.

Table 4 compares this one-round model with the Round 4 model from the default setup and the Round 8 model from the fewer-interactions-per-round setup, both from Section 6.2. Overall, the one-round model does worse on F1 score, illustrating the benefit of the multi-round design. The higher F1 score on the answerable set the one-round model shows is explained by the significant under-prediction of questions as "unanswerable".

## 6.4 Sensitivity Analysis

We characterize the sensitivity of the bandit learning process through controlling the set of feedback data and the set of initial models.

**Sensitivity Analysis of Round 1 Models** Analyzing model sensitivity over multiple rounds of interaction is prohibitively costly. We focus on analyzing the sensitivity of the model at Round 1 relatively to the set of feedback examples it is being trained on. We collect 10 different sets of 200 examples and compare the learning outcomes.

Figure 4 shows the performance of models learned from those 10 different sets. The overall F1 score, F1 on the answerable subset, and classification accuracy present relatively smaller variance (standard deviation $\sigma$ 3.32, 2.89, and 2.44) on the full test set from Section 6.1. We observe a larger variance in the F1 on the unanswerable subset and in the percentage of predicted unanswerable examples ($\sigma$ 11.51 and 9.55). This variance may be due to the challenge of improving the binary classification head on unbalanced data: the percentage of model-predicted unanswerable examples in the feedback data is 7% on average across 10 different sets during the first round.

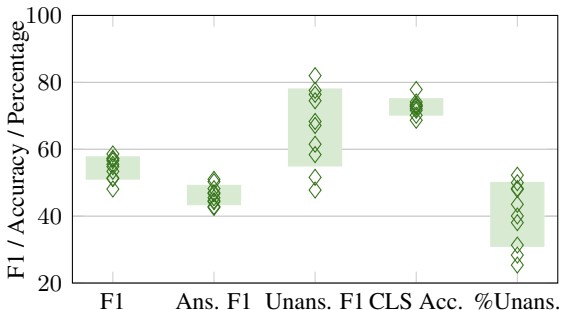

Figure 4: Performance of Round 1 models learned from 10 different sets of feedback examples on the full test set: F1, F1 on the ground truth answerable subset (Ans. F1), F1 on the ground truth unanswerable subset (Unans. F1), classification accuracy (CLS Acc.), and percentage of predicted unanswerable outputs (%Unans.). Bars represent variance, and are centered at the mean value.

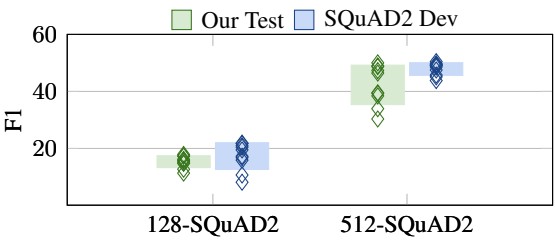

Figure 5: Performance of initial models trained on 10 different sets of 128 or 512 SQuAD2 examples on our full test set and the SQuAD2 development set. Bars represent variance, and are centered at the mean value.

**Sensitivity Analysis of Initial Models**    The initial models used in our study are initialized using either 512 or 128 supervised examples from SQuAD2 dataset. We analyze the variance they may present. We randomly sample different sets of supervised examples from the SQuAD2 training data to study the sensitivity of the initial model to its supervised training data.

Figure 5 shows the performance of initial models trained on 10 different sets of 512 or 128 SQuAD2 examples on our full test set from Section 6.1 and SQuAD2 development set. On SQuAD2 development set, 512-SQuAD2 models show a smaller variance in F1 than 128-SQuAD2 models (standard deviation $\sigma$ 2.2 vs 4.7). On our test set, we observe a different result: 512-SQuAD2 models have a larger variance in F1 than 128-SQuAD2 models ($\sigma$ 6.9 vs 2.1). We find that the larger variance in 512-SQuAD2 model performance on our test set is due to the large variance in F1 on the ground-truth unanswerable subset: $\sigma$ 24.4. In contrast, F1 on the ground-truth unanswerable subset for 128-SQuAD2 models have standard deviation 6.4, but with a much lower mean (13.6 vs 38.6). 128-

SQuAD2 models on average only predict 11.9% unanswerable outputs on our test set.

## 7 Conclusion

We deploy an extractive QA system, and improve it by learning from human user feedback, observing strong performance gains over time. Our experiments show such continual improvement of systems is compelling for domain adaptation, providing a practical avenue for practitioners to bootstrap NLP systems from existing academic datasets, even when the domain gap is relatively large. We also study the effects of the update schedule (i.e., how frequently to train and re-deploy a new model) and model design (i.e., by using a separate answerability classification head). Through our focus on a lexically-rich user-facing scenario, we hope our study will inspire further research of NLP problems in interactive scenarios and the learning potential user interactions provide. Important directions for future work include learning from other, more complex signals (e.g., natural language feedback), studying non-extractive scenarios, and developing robustness to adversarial behaviors (e.g., attempts to sabotage the system through feedback).

## Limitations

We simulate information-seeking users with paid crowdworkers. This has limitations as far as reflecting the behaviors and incentives of real users. For example, it is not clear how often real users will provide feedback at the end of the interaction, or how much effort they will put into validating the system output. On the other hand, crowdworkers have less incentive to design their questions so that they will be able to get a satisfying answers, as opposed to real users who likely require the answers to achieve their intent. Also, changes in user population and preferences are unlikely to be reflected in crowdworkers behavior.

We only elicit cooperative interactions in our experiments. Our approach does not address potential adversarial behavior, implicitly assuming collaborative users. However, real-life scenarios are unlikely to allow such assumptions. In general, adversarial user behavior forms a security risk for learners that rely on user feedback, and even more when user input is part of the process, allowing users stronger control on how they may adversarially attempt to steer the system. This aspect of NLP systems is currently not broadly studied. We

hope our work will enable further work into this important problem.

Other aspects constraining the generality of our conclusions, but are not strictly limitations of our specific work, are our focus on extractive QA, where the output is relatively easy to visualize for validation, and our simple feedback mechanism, which is not as information rich as other potential modes of feedback (e.g., natural language). We hope our work will serve as a catalyst for studies that cover these important problems.

## Legal and Ethical Considerations

Section 5 and Appendix A detail our experimental setup, hyperparameter search, and computational budget to support reproducibility. Our data and codebase are released at `https://github.com/lil-lab/qa-from-hf`. SQuAD2 and TyDiQA that we use for initial training and evaluation are publicly available datasets from prior work.

The content shown to crowdworkers in our studies is from Wikipedia articles, so we expect crowdworkers participating in our task to be free from exposure to harmful content and the content to not pose intellectual property issues. While our data does not contain harmful content, we cannot predict its impact when it or our models are used beyond research context similar to ours. We urge such uses to be accompanied by careful analysis and appropriate evaluation.

In general, learning from user feedback, especially on user-generated data, poses risks for diverting the system behavior in unwelcome directions. Identifying and mitigating such risks are open and important problems for future work to study.

## Acknowledgements

This research was supported by Open Philanthropy, a Google Faculty Research Award, and NSF under grant No. 1750499. We thank participating crowdworkers on Amazon Mechanical Turk for their contributions, and hope they enjoyed interacting with our system. We thank Kianté Brantley, Jonathan D. Chang, Justin Chiu, Daniel D. Lee, and Travers Rhodes for helpful discussions. We thank Justin Chiu and Michael J.Q. Zhang for their valuable feedback on our drafts. We thank members of Cornell NLP and UT Austin NLP community for providing feedback.

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

## A  Training Details

**Training Initial Models**  We randomly sample 512 training examples from the SQuAD2 training set and use a learning rate of 3e-5 with a linear schedule, batch size 10, and 10 epochs. The loss terms for span prediction and answerability classification are simply added. We use the same hyperparmeter setup for training the initial model on 128 SQuAD2 examples except that we add a coefficient of 10 to the answerability classification loss. For the initial model trained on NewsQA, we use a learning rate 2e-5 with a linear schedule, batch size 40, and 4 epochs.

**Bandit Learning**  We use batch size 35, and perform hyperparameter search with a linear schedule of learning rates (5e-5, 3e-5, 1e-5, 9e-6, 7e-6, 5e-6, 3e-6), number of epochs (14, 20, 24, 30), and entropy penalty coefficients $\gamma$ (1.5, 1.7, 1.9, 2.0, 2.1, 2.3, 2.5, 3.0, 5.0). We pick the checkpoint with highest F1 on our development set, a set of 402 examples annotated during early pilot studies. At each round, these hyperparameter searching experiments take around 105h on one NVIDIA GeForce RTX 2080 Ti.

## B  User Study Setup

We design an interface where users ask information-seeking questions and provide feedback to model-predicted answers to their own questions.[11] Figure 8 shows screenshots of our interface. We use this interface throughout our studies.

We familiarize workers with our task and criteria using detailed guidelines and examples of reasonable questions and user feedback. Before doing our tasks, workers must pass a qualification process by correctly answering at least 9/10 questions regarding the task. The examples we use to guide workers and the qualification tests for both the feedback and test data annotation tasks are available at https://github.com/lil-lab/qa-from-hf/tree/main/data-collection.

**Question Writing Guidelines**  We encourage workers to ask information-seeking questions by having them pick a topic they find interesting among four topic randomly selected from a pool, and then choose one aspect out of four regarding the chosen topic. For topic selection, we provide a

---

[11]We customize a UI design from CodingNepal on "Create a Quiz App with Timer using HTML CSS & JavaScript".

---

short introduction to each topic. After selecting a topic and an aspect, workers are instructed to ask a relevant question that they do not have an answer to. We explicitly guide workers to avoid yes-or-no questions and questions that obviously cannot be answered by a short single sentence.

**Feedback Guidelines**  We explicitly guide workers to read the context paragraph in full to give feedback when the model returns "unanswerable", and to give "partially correct" feedback if the answer contains irrelevant information. Beyond that, we guide workers through examples that express the following guidelines:

- "Correct" if the answer highlighted in the context paragraph correctly answers the question, or points out that the question cannot be answered by a single span in the paragraph.
- "Partially Correct" if the given answer only partially answers the question.
- "Wrong" if the given answer highlighted in the paragraph does not answer the question at all, or if the output is "unanswerable given the paragraph" when the question can be answered by a span in the paragraph.

**Test Data Annotation**  We collect test examples with ground-truth answer annotation. We ask workers to annotate answers to their questions instead of providing feedback to model-predicted answers. Workers are instructed to either select a single span in the paragraph as an answer or indicate that the question is unanswerable given the context paragraph. We intentionally combine this annotation task with the main feedback task, so that workers do not know whether they will provide annotation to their own question or provide feedback to a model answer. In order to have three annotations per test example, we design an additional task to collect two extra annotations per test example: workers are given a question and context paragraph, and provide answer annotation. We qualify workers by providing examples of reasonable annotations and only work with workers who correctly pass a qualifier composed of three questions.

During early pilot studies, we collect a small set (402 examples) of annotated examples. We use this set for hyperparameter tuning during development.

**Worker and Study Details**  For both feedback task and annotation task, we work with turkers that have a HIT (Human Intelligence Task) rate greater than 98% with at least 500 completed HITs.

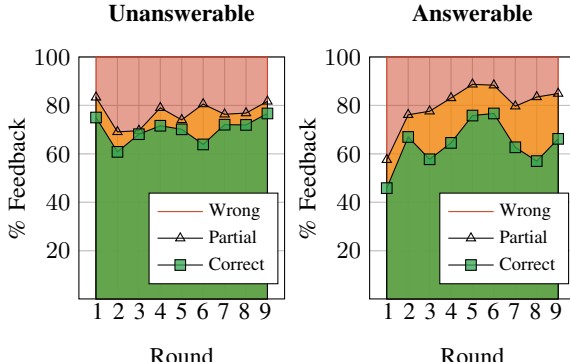

Figure 6: Percentage of user feedback received at every round during the long-term experiment. Left column shows the subset where the model predicts unanswerable, and the right column shows the complement.

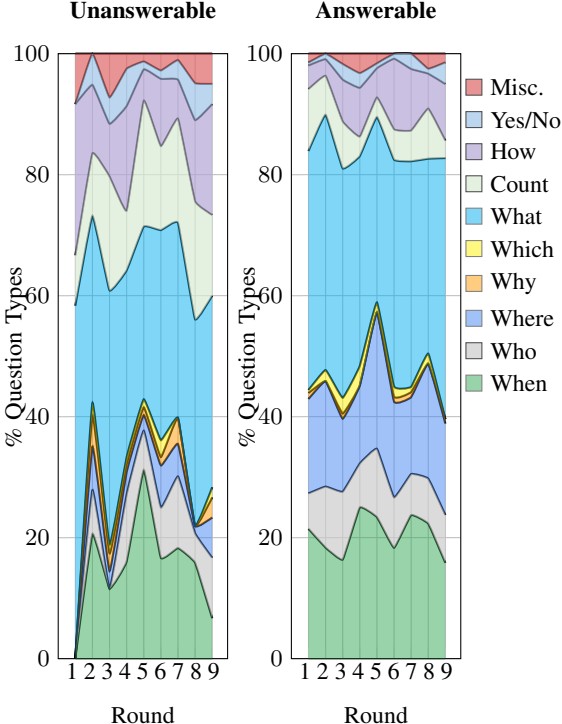

Figure 7: Statistics of question types per round in the long-term experiment. Left column shows the subset where the model predicts unanswerable, and the right column shows the complement.

Across all experiments, 103 turkers have passed our qualification test and participated in our tasks. No turker has done more than 5% of the HITs in total. We pay $0.65 USD for each feedback example and $0.4 USD for each additional annotation. The estimated hourly pay for both tasks are $15 USD. Our user study spans from February 8th to April 25th, 2023.

| | |
|---|---|
| Question: How large was the sea at its peak volume? | |
| Answer: [Unanswerable given the context paragraph] | |
| Context: The Amu Darya river flowed into the Caspian Sea via the Uzboy channel until the Holocene. Geographer Nick Middleton believes it did not begin to flow into the Aral Sea until that time. | |
| Feedback: Correct | |

Question: Who was the original inventor or programmer of Solaris?
Answer: [Unanswerable given the context paragraph]
Context: In 1987, AT&T Corporation and Sun announced that they were collaborating on a project to merge the most popular Unix variants on the market at that time ... On September 4, 1991, Sun announced that it would replace its existing BSD-derived Unix, SunOS 4, with one based on SVR4. This was identified internally as SunOS 5, but a new marketing name was introduced at the same time: Solaris 2 ...
Feedback: Partially Correct

Question: Who was the person most suspected of being Jack the Ripper?
Answer: an educated upper-class man, possibly a doctor
Context: The concentration of the killings around weekends and public holidays and within a short distance of each other has indicated to many that the Ripper was in regular employment and lived locally. Others have opined that the killer was an educated upper-class man, possibly a doctor or an aristocrat who ventured into Whitechapel from a more well-to-do area. Such theories draw on cultural perceptions such as ...
Feedback: Partially Correct

Question: In which state is Ulm located in?
Answer: Bavaria
Context: Ulm lies at the point where the rivers Blau and Iller join the Danube, at an altitude of 479 m (1,571.52 ft) above sea level. Most parts of the city, including the old town, are situated on the left bank of the Danube; only the districts of Wiblingen, Gögglingen, Donaustetten and Unterweiler lie on the right bank. Across from the old town, on the other side of the river, lies the twin city of Neu-Ulm in the state of Bavaria, smaller than Ulm and, until 1810, a part of it ...
Feedback: Wrong

Table 5: User interaction examples: each example is composed of user question, context paragraph, model-predicted answer, and user feedback.

## C Data in Long-Term Experiment

**Interaction Examples** Table 5 lists additional interaction examples from our long-term study (Section 6.1).

**User Feedback** Figure 6 visualizes the distribution of user feedback at every round, partitioned based on whether the model predicts unanswerable or an answer span. The percentage of "correct" feedback is increasing on both subsets, illustrating

that the model is improving over time.

**QA Data**   Figure 7 visualizes the distribution of question types, partitioned based on whether the model predicts unanswerable or an answer span. The distribution over question types varies at each round, with the majority of questions starting with "what" and "when". The model mostly predicts a span in the context to answer questions regarding "where", while it chooses the unanswerable option more often towards counting-related questions.

## Instructions

★ This task includes 4 steps:
    1. Choose a topic that interests you from a list of topics. We provide a brief intro to each topic.
    2. Select one aspect of the chosen topic to ask a question about.
    3. Ask a question that you are curious to know more about. The question must be about the topic and aspect you selected.
    4. We will generate an answer to your question, and highlight it in a paragraph, or show "[Unanswerable given the paragraph below]". You will provide feedback on our response given the paragraph (correct, partially correct or incorrect.).

★ When asking your question, it is critical that:
    ☆ You ask a question that can be answered by a short single sentence. We can only answer such questions.
    ☆ Do NOT ask a yes-or-no question.
    ☆ Put a question mark at the end of your question. Otherwise, the button to the next step will not show up.

★ When providing feedback to the given answer, keep in mind that:
    ☆ We may return "[Unanswerable given the paragraph below]".
    ☆ If the given answer is "unanswerable", please read the full paragraph to make sure that this paragraph cannot answer your question. If you find a correct answer in the paragraph, please indicate our no-answer output as "incorrect".
    ☆ If the given answer could answer your question but contains irrelevant content, label it as "partially correct".

★ Please read our example google doc.

★ First-time users need to successfully complete a qualifier questionnaire before starting the task.

★ Keyboard shortcuts:
    ☆ 1, 2, 3, 4: numbered options in each single selection
    ☆ Enter: go to the next step, or submit after completing all steps

★ Please read our consent form before starting the task. By continuing with this task, you acknowledge that you understand our consent form, and agree to take part in this research.

Start ↵

## Topic Selection

### Select a topic that interests you:

| | |
|---|---|
| 1. Wrigley Field | Wrigley Field is a Major League Baseball (MLB) stadium located on the North Side of Chicago, Illinois. It is the home of the Chic... |
| 2. Lighthouse of Alexandria | The Lighthouse of Alexandria, sometimes called the Pharos of Alexandria (; Ancient Greek: ὁ Φάρος τῆς Ἀλεξανδρείας, contemporary K... |
| 3. Andrew Cuomo | Andrew Mark Cuomo ( KWOH-moh; Italian: [ˈkwɔːmo]; born December 6, 1957) is an American lawyer and politician who served as the 56... |
| 4. Battle of Okinawa | The Battle of Okinawa (Japanese: 沖縄戦, Hepburn: Okinawa-sen), codenamed Operation Iceberg,:17 was a major battle of the Pacific W... |

Optional discussion forum: **Discord**

Instructions

## Aspect Selection

**If needed, below is a brief intro to** Wrigley Field **:**

Wrigley Field is a Major League Baseball (MLB) stadium located on the North Side of Chicago, Illinois. It is the home of the Chicago Cubs, one of the city's two MLB franchises. It first opened in 1914 as Weeghman Park for Charles Weeghman's Chicago Whales of the Federal League, which folded after the 1915 baseball season. The Cubs played their first home game at the park on April 20, 1916, defeating the Cincinnati Reds 7–6 in 11 innings. Chewing gum magna...

**Select an aspect that interests you:**

1. Accessibility and transportation

2. History

3. Commemorative stamps

4. Features

Optional discussion forum: Discord

Instructions    Restart

## Question

**If needed, below is a brief intro to** Wrigley Field **:**

Wrigley Field is a Major League Baseball (MLB) stadium located on the North Side of Chicago, Illinois. It is the home of the Chicago Cubs, one of the city's two MLB franchises. It first opened in 1914 as Weeghman Park for Charles Weeghman's Chicago Whales of the Federal League, which folded after the 1915 baseball season. The Cubs played their first home game at the park on April 20, 1916, defeating the Cincinnati Reds 7–6 in 11 innings. Chewing gum magna...

**Ask a question about** History **:**

When was Wrigley Field built?

Optional discussion forum: Discord

Instructions    Restart    Next Step ↵

## Feedback

### Question:

When was Wrigley Field built?

### Answer:

April 23, 1914

### Context:

Baseball executive Charles Weeghman hired his architect Zachary Taylor Davis to design the park, which was ready for baseball by the home opener on April 23, 1914. The original tenants, the Chicago Whales (also called the Chi-Feds), came in second in the Federal League rankings in 1914, and won the league championship in 1915. In late 1915, Weeghman's Federal League folded. The resourceful Weeghman formed a syndicate including the chewing gum manufacturer William Wrigley Jr. to buy the Chicago Cubs from Charles P. Taft for about $500,000. Weeghman immediately moved the Cubs from the dilapidated West Side Grounds to his two-year-old park. In 1918, Wrigley acquired the controlling interest in the club. In November 1926, he renamed the park Wrigley Field. In 1927, an upper deck was added, and in 1937, Bill Veeck, the son of the club president, planted ivy vines against the outfield walls after seeing the ivy planted at Perry Stadium, Indianapolis.

### How do you evaluate the highlighted answer?

1. Correct

2. Partially correct

3. Wrong

Optional discussion forum: Discord (Feel free to ask questions regarding your HIT.)

### Optional comment:

Any comments or feedback about this task? Type here...

Instructions

Figure 8: Snapshots of our interface for feedback task.