# OpenReview forum: "Continually Improving Extractive QA via Human Feedback"
_EMNLP/2023/Conference — EMNLP 2023 Main_

### Official Review · Reviewer_Qenp · 2023-08-05

**Soundness:** 4

**Excitement:**

4: Strong: This paper deepens the understanding of some phenomenon or lowers the barriers to an existing research direction.

**Paper Topic And Main Contributions:**

The paper focuses on extractive QA. Specifically, the paper proposes an approach to

**Questions For The Authors:**

A. What is the advantage of improving using human feedback compared to using human labels to continuously train the model? It is easier and cheaper to collect? Or other advantages?

**Reasons To Accept:**

The idea of improving QA with three-option human feedback is nice, which shed lights on iteratively improving deployed QA systems. I am not an expert on reinforce learning, but the methodology is clear and able to follow. From the experimental results, the improvements are significant. The paper also provide comprehensive discussions on variants of experimental settings.

**Reasons To Reject:**

The motivation of improving extractive QA with human feedback is not clearly explained. By reading the introduction, I don't have a clear sense of why we need human feedback for QA.

**Reproducibility:**

4: Could mostly reproduce the results, but there may be some variation because of sample variance or minor variations in their interpretation of the protocol or method.

**Reviewer Confidence:**

3: Pretty sure, but there's a chance I missed something. Although I have a good feel for this area in general, I did not carefully check the paper's details, e.g., the math, experimental design, or novelty.

---

> ### Author Rebuttal · Authors · 2023-08-28
>
> Thanks for your review.
>
> User feedback is fundamentally different from labeled data (i.e., as obtained from annotators). First, it doesn’t cost money (so: free!), but arises naturally from the interaction of users with the system. We paid workers to take the place of users, but this is just to run experiments. Second, it’s much more abundant, because it arises from users interacting with the system, and pools of users are generally orders of magnitude larger than pools of annotators. Third, while annotated data reflects an assumption about a static data distribution, learning from feedback over the system’s lifetime enables taking into account adapting over time to evolving data distribution, both as users change their behavior and the world changes (i.e., the answers to questions change – see Zhang and Choi EMNLP 2021). So, while it creates a much more challenging learning scenario (which we show our method effectively addresses), it is across the board much better as a learning signal (once you can show that you can use it effectively, which we show). We discussed this motivation in the introduction, but will further improve it.

---

### Official Review · Reviewer_CpKn · 2023-08-06

**Soundness:** 3

**Excitement:**

4: Strong: This paper deepens the understanding of some phenomenon or lowers the barriers to an existing research direction.

**Paper Topic And Main Contributions:**

The authors propose a method to continually improve extractive QA via human feedback. The method does have high practical value and may be applied in building other NLP application systems. Extensive and carefully designed experiments demonstrate that an extractive QA system can be improved via interaction with human users over time and further broaden our understanding about continual learning with user feedback.

**Questions For The Authors:**

1. What's the possible performance upper bound of the proposed method? How many rounds of interactions are optimal in general?

**Reasons To Accept:**

1. The authors propose a method to continually improve extractive QA via human feedback. The proposed method does have high practical value and may be applied in building other NLP application systems.

2. Extensive and carefully designed experiments demonstrate that an extractive QA system can be improved via interaction with human users over time.

**Reasons To Reject:**

1. Uncertainties introduced by human interactions are likely to affect the results and need to be carefully controlled with a well-defined guideline.

**Reproducibility:**

4: Could mostly reproduce the results, but there may be some variation because of sample variance or minor variations in their interpretation of the protocol or method.

**Reviewer Confidence:**

4: Quite sure. I tried to check the important points carefully. It's unlikely, though conceivable, that I missed something that should affect my ratings.

**Typos Grammar Style And Presentation Improvements:**

Typos or grammar errors:

1. line 345, "We are focused on ..."
2. line 369, "the systems sees"
3. line 481, "does not an explain ..."
4. line 555, "the that"
5. line 620, "out approach"

---

> ### Author Rebuttal · Authors · 2023-08-28
>
> Thanks for your review.
>
> Regarding the reproducibility of our experiments: our code and data will be made publicly available. They will enable others to easily reproduce the experiments reported in the paper. We agree that working with human users introduces uncertainties to reproduction. This is a general challenge when working with humans (same for human evaluation), but there are no substitutes for working with humans for both learning from human feedback (as we do) and human evaluation. This is a known research challenge in this area, but does not detract from how important/necessary these problems are and how excited the community is about them. Even more important, we actually worked really hard to show how reproducible our experiments are, and directly show that our work is robust to this uncertainty. This is shown by our reproduction of the default setup in a separate experiment. Our reproduction (Section 6.2 and default setup plot in Figure 3) shows similar improvements trends to the long-term experiment (Section 6.1 and Figure 2). The cost and user study details are clearly reported in the main paper and Appendix B to further support the reproducibility of our work.
>
> Regarding the question on the possible performance upper bound: our goal was to show the feasibility and effectiveness of learning from feedback with a straightforward approach. The upper bound question is interesting, but complex and likely doesn’t have a clear answer. This is because of the constant distribution shift. Throughout the lifetime of the system, and as it learns, there are two major causes for constant data distribution shift: (a) the world changes; and (b) the users change their behavior in response to the system changing its behavior or as new users start using the system. In this sense, this scenario reflects a multi-agent system (with the system being one agent, and the human users as many other agents). This means that even if performance doesn’t change much after a while (something that may happen), the adaptation doesn’t stop. So this process should continue throughout the system’s lifetime. Further study of this interesting problem is something that our work enables, as the first to show such a system improving over time. We will discuss this in the paper.

---

### Official Review · Reviewer_4d65 · 2023-08-11

**Soundness:** 4

**Excitement:**

4: Strong: This paper deepens the understanding of some phenomenon or lowers the barriers to an existing research direction.

**Paper Topic And Main Contributions:**

The paper is about a reinforcement learning for extractive QA. authors trained a model in different rounds using human feedback to improve the model. They also tested different version of the model with different initialization and on different dataset to ensure the performances on different domain. Authors relies on crowdworkers for the feedback. Performances show the effectiveness of the approach.

**Questions For The Authors:**

1 Can you better clarify how you can ensure feedback quality?
2 Why you did not perform an evaluation of the annotators job?

**Reasons To Accept:**

+ well written paper
+ novel approach to improve extractive QA
+ relevant results and a new interesting research for qa systems
+ good experiments and ablations

**Reasons To Reject:**

+ miss a proper evaluation of the quality of the crowdworkers to avoid misleading errors
+ no clear how ensure feedback quality

**Reproducibility:**

4: Could mostly reproduce the results, but there may be some variation because of sample variance or minor variations in their interpretation of the protocol or method.

**Reviewer Confidence:**

4: Quite sure. I tried to check the important points carefully. It's unlikely, though conceivable, that I missed something that should affect my ratings.

---

> ### Author Rebuttal · Authors · 2023-08-28
>
> Thanks for your review. We evaluated feedback quality, as described in Footnote 7. Overall, our feedback data was relatively clean (18% noise rate). We will move this content to the main text and discuss it further. We used a qualifier and MTurk criteria (e.g., 98% acceptance rate) to filter spamy workers, and make sure workers understand the system and are well qualified to interact with it (i.e., that they are similar to users in this respect). While this noise rate may seem high, our method is robust to it, which is the important thing. Equally important, our feedback providers are not annotators, but take the place of users. The feedback comes to reflect naturally occuring feedback. So training our workers (beyond what we did) to give perfect feedback is actually not desirable, because you wouldn’t train your users when deploying a system. We agree there are likely interface design choices that can influence feedback quality, and we believe this is an important direction for future work in the intersection of HCI and NLP that our work enables. Another exciting option for future work is thinking of methods to filter bad feedback. Our work doesn’t come to establish the ultimate method. It comes to show effectiveness and feasibility, and on both accounts we believe our results are extremely strong (all that with real human feedback!). There is only so much that can come within the scope of an ACL paper, and our experimental costs are already very high because each experiment requires human work. As the reviewer notes, the scope of this area of research is huge, and there are so many exciting questions. We hope the reviewers agree that it’s a strength of our work that it outlines so many exciting directions for future work.

---

### Meta-Review · Area_Chair_Py3U · 2023-09-18

**Recommendation:** 4

**Metareview:**

This paper describes a pipeline to improve an extractive QA system using human feedback loop. The proposed approach is not only feasible for an extractive QA system, but has potential to be extended to other NLP applications. Reviewers appreciated the well-strutted content, innovative methods, and comprehensive experiments.

The questions raised by reviewers including the motivation of using human feedback vs. human labels, the quality of human feedback, and uncertainties of human feedback, are appropriately addressed by the authors. After the rebuttal period, the authors mentioned releasing the annotated dataset and annotation guidelines, which would enhance reproducibility.

---

### Decision · Program_Chairs · 2023-10-07

**Decision:**

Accept-Main

**Comment:**

This paper describes a pipeline to improve an extractive QA system using human feedback loop. The proposed approach is not only feasible for an extractive QA system, but has potential to be extended to other NLP applications. Reviewers appreciated the well-strutted content, innovative methods, and comprehensive experiments.

The questions raised by reviewers including the motivation of using human feedback vs. human labels, the quality of human feedback, and uncertainties of human feedback, are appropriately addressed by the authors. After the rebuttal period, the authors mentioned releasing the annotated dataset and annotation guidelines, which would enhance reproducibility.